# Long-Term Treatment of Cancer-Associated Thrombosis (CAT) Beyond 6 Months in the Medical Practice: USCAT, a 432-Patient Retrospective Non-Interventional Study

**DOI:** 10.3390/cancers12082256

**Published:** 2020-08-12

**Authors:** Isabelle Mahé, Ludovic Plaisance, Céline Chapelle, Silvy Laporte, Benjamin Planquette, Laurent Bertoletti, Francis Couturaud, Nicolas Falvo, Lionel Falchero, Isild Mahé, Hélène Helfer, Jean Chidiac, Guy Meyer

**Affiliations:** 1Hôpital Louis Mourier, Assistance Publique des Hôpitaux de Paris, Université de Paris, 92700 Colombes, France; ludovic.plaisance@aphp.fr (L.P.); isild.mahe@sfr.fr (I.M.); helene.helfer@aphp.fr (H.H.); jean.chidiac@aphp.fr (J.C.); 2F-CRIN INNOVTE network, F-42055 Saint-Etienne, France; silvy.laporte@chu-st-etienne.fr (S.L.); benjamin.planquette@aphp.fr (B.P.); laurent.bertoletti@gmail.com (L.B.); francis.couturaud@chu-brest.fr (F.C.); nicolas.falvo@chu-dijon.fr (N.F.); guy.meyer@aphp.fr (G.M.); 3Innovative Therapies in Haemostasis, Institut National de la Sante et de la Recherche Medicale (INSERM), Université de Paris, F-75006 Paris, France; 4Université Jean Monnet, Université de Lyon, SAINBIOSE INSERM U1059, F-42023 Saint-Etienne, France; celine.chapelle@chu-st-etienne.fr; 5Unité de Recherche Clinique, Innovation, Pharmacologie, Centre Hospitalo-Universitaire Saint-Etienne, F-42055 Saint-Etienne, France; 6Hôpital Européen Georges Pompidou, Assistance Publique des Hôpitaux de Paris, Université de Paris, 75015 Paris, France; 7Centre Hospitalo-Universitaire de St-Etienne, Service de Médecine Vasculaire et Thérapeutique, F-42055 Saint-Etienne, France; 8Equipe Dysfonction Vasculaire et Hémostase, Université Jean-Monnet, INSERM, UMR1059, F-42023 Saint-Etienne, France; 9Centre Hospitalo-Universitaire de Saint-Etienne, INSERM, CIC-1408, F-42055 Saint-Etienne, France; 10Hôpital de la Cavale Blanche CHRU de Brest, Centre Hospitalo-Universitaire de Brest, CIC INSERM 1412, EA 3878 (GETBO), 29609 Brest, France; 11Service de Médecine Interne, Centre Hospitalo-Universitaire de Dijon, 21000 Dijon, France; 12Hôpital Nord-Ouest Villefranche-sur-Saône, 69400 Gleize, France; LFalchero@lhopitalnordouest.fr

**Keywords:** cancer, thrombosis, anticoagulants, bleeding, death

## Abstract

Background: extended anticoagulant therapy beyond the initial 6 months is suggested in patients with cancer-associated thrombosis (CAT) and active cancer. Few data are available on patient management and outcomes on the period between 6 and 12 months after the venous thromboembolism (VTE) event. Objectives: our objective was to document patient management and outcomes beyond 6 months and up to 12 months in CAT patients initially treated for 6 months with tinzaparin. Methods: adult CAT patients with a cancer still alive at the end of an initial 6-month treatment period were eligible to participate in this retrospective non-interventional French multicenter study. Results: a total of 432 patients aged 66.5 ± 12.7 years were available to participate in this study. Out of the patients included in the study, the anticoagulant treatment was maintained in 348 of 422 documented patients (82.5%) while it was discontinued in 74 (17.5%) patients (before the end or at the end of the initial 6-month treatment period). Between 6 and 12 months, 24 patients (5.7%) experienced VTE recurrence, while 21 (5.1%) patients had clinically relevant bleeding, 11 patients (2.7%) had major bleeding and 96 patients (22.3%) died, mostly from cancer. VTE recurrence was more frequent in patients with lung (14.3%) and colorectal cancer (6.0%) while major bleeding was more frequent in patients with colorectal cancer (6.0%). Conclusion: clinical outcomes were consistent with previous observations and variable according to the type of cancer. Further clinical research is required to orient the management of patients with CAT beyond 6 months based on cancer-specific treatment strategies.

## 1. Introduction

The risk of venous thromboembolism (VTE) in patients with cancer is 7-fold higher compared to patients without cancer [1]. Managing patients with cancer-associated thrombosis (CAT) represents a significant challenge since they are at higher risk of both VTE recurrence and major bleeding compared to patients without cancer [2,3].

Clinical practice guidelines recommend a duration minimum of 3 to 6 months anticoagulant treatment [4,5,6,7,8,9,10]. However, there is no established consensus on the optimal duration of the anticoagulant treatment in patients with CAT, especially beyond 6 months. Most guidelines tend to recommend the extension of the anticoagulant therapy for as long as the cancer is active and/or the patient receives an antineoplastic treatment which may be associated with an increased risk of VTE recurrence, although these recommendations are not based on randomized trials (Table 1). The decision about the therapeutic strategy beyond 6 months is therefore made on a case by case based on the expected benefit–risk balance.

Several studies, mostly uncontrolled, have evaluated extended anticoagulant therapy for patients with CAT, suggesting that long-term anticoagulant treatment beyond 6 months may be associated with a lower risk of VTE recurrence and bleeding compared to the initial 6-month treatment period [11,12,13,14]. However, the studies available to date provide limited orientation for the management of CAT patients beyond 6 months after the index VTE mainly due to the relatively small patient sample size.

Two large prospective cohort studies have documented the treatment of CAT patients with tinzaparin for 6 months. The PREDICARE study included 409 patients of whom 28 patients had a recurrent VTE, yielding a cumulative incidence of 7.3% (95% CI: 4.9–11.1) while 15 patients had a major bleeding yielding a cumulative incidence of 3.7% (95% CI: 2.3%; 6.0%) during the 6-month treatment period [15]. A total of 310 CAT patients with objectively diagnosed VTE were enrolled to receive 6-month anticoagulant therapy with tinzaparin in the aXa study (NCT02898051; https://clinicaltrials.gov/ct2/show/NCT02898051).

The objective of this observational study was to briefly describe in real-world practice the use of the anticoagulant treatment beyond 6 months and up to 12 months following index VTE and to document clinical outcomes i.e., VTE recurrence, bleeding and deaths in CAT patients initially treated for 6 months in both PREDICARE and aXa studies.

## 2. Material and Methods

### 2.1. Inclusion Criteria

Adult patients with cancer and objectively diagnosed acute VTE previously included in both prospective observational cohort studies, aXa and PREDICARE, and who were still alive at the end of the initial 6-month treatment period with tinzaparin, and having given their consent for the use of their data were eligible to participate in Usual Care of Cancer Associated Thrombosis (USCAT), a retrospective non-interventional multicenter cohort study. The clinical protocol was submitted and approved by the Institutional Review Board of the University Hospital of Saint-Etienne, France (Institutional Review Board: IORG0007394, Number RBN342018/CHUSTE).

### 2.2. Study Outcomes

Main study outcome was the description of the anticoagulant treatment for the management of CAT patients from the 6th month to the 12th month following the index VTE.

Secondary outcomes included: (1) the incidence of recurrent VTE events confirmed by appropriate imaging studies in any venous or pulmonary arterial circulation including pulmonary embolism (PE), PE-related death, deep-vein thrombosis (DVT); (2) the incidence of clinically relevant bleeding (CRB), defined as the sum of major bleeding and clinically relevant non-major bleeding (CRNMB) as per the International Society on Thrombosis and Haemostasis classifications [16]. Major bleeding events included fatal bleeding, overt bleeding in a critical area or organ, and bleeding causing a fall in hemoglobin level of 20 g/L or more or leading to transfusion of 2 or more units of whole blood or red blood cells. Clinically relevant non major bleeding included bleeding events that did not meet the criteria for major bleeding but did require a medical intervention by a healthcare professional, lead to hospitalization or increased level of care, or prompted a medical appointment [17]; and, (3) incidence of all-cause mortality.

### 2.3. Data Management and Statistics

The same independent investigator had access to the primary data and reviewed medical records of all included patients and recorded data in a standardized case report form. Relevant data, including patient demographics, cancer status and treatment, anticoagulant therapy, index VTE events, VTE recurrences, bleeding events and deaths were collected up to 12 months after the index VTE event. Information on changes in cancer status, cancer treatment and anticoagulant therapy was collected.

A sample size of 250 to 300 patients was considered appropriate to allow analyses of 6-to-12-month data following the index VTE based on descriptive statistics.

The cumulative incidence of VTE recurrences and bleeding events was estimated by the Kalbfleisch and Prentice method to take in account the competing risk of death and the cumulative incidence of deaths was estimated by the Kaplan Meier method. The Fine and Gray regression model taking in account competing risks was used to identify potential risk factors of VTE recurrence and bleeding based on univariate and multivariate analyses. Statistical analyses were performed using SAS version 9.4 software (SAS Institute Inc, Cary, NC, USA).

## 3. Results

Between 4 August 2011 and 21 April 2016, 719 patients were included in aXa and PREDICARE studies in 59 French centers, of which 432 participated in the long-term follow-up USCAT study (Figure 1). Investigators and study centers are listed in the Appendix A. Patients’ characteristics at inclusion are summarized in Table 2. The mean age of the study population was 66.5 ± 12.7 years while 29.6% were aged 75 years or more. Most patients (91.9%) had one single cancer site. It was a solid tumor in 91.2% of cases of which 21.6%, 20.1% and 16.8% were colorectal, lung and breast cancers, respectively. Tumor was hematologic in 6.9% of patients. The index VTE diagnosis was a PE in 318 patients (73.6%) either isolated (233 patients) or combined with deep vein thrombosis (DVT, 84 patients) and DVT alone in 114 patients (26.4%).

### 3.1. Maintained Anticoagulant Treatment Beyond 6 Months Following the Index VTE

At 6 months following the index VTE, the anticoagulant treatment was maintained in 348 patients (82.5%) while it had been discontinued in 74 patients (17.5%) of the 422/432 documented patients.

Ongoing anticoagulant treatment at 6 months included low-molecular-weight heparin (LMWH), vitamin K antagonists (VKA) and direct oral anticoagulant (DOAC) in 256 (73.6%), 56 (16.1%) and 30 patients (8.6%), respectively, while unfractionated heparin and fondaparinux were each prescribed in three patients (0.9%).

The median treatment duration with LMWH was 137 days, resulting in a median total treatment duration since the index VTE of 292 days. The median treatment duration with VKA and DOACs was 183 days, resulting in a median total treatment duration since the index event of 189 days and 183 days resulting in a median total treatment duration since the index event of 184 days, respectively.

LMWHs were definitively discontinued in 86 patients (33.7%); besides death, the main reasons were a favorable course of the cancer (16 patients, 18.6%) and VTE disease (11 patients, 12.8%), whereas concern about bleeding risk was low (2 patients, 2.3%). The time to definitive discontinuation was between 50 and 100 days beyond 6 months.

LMWHs were switched to oral anticoagulants (DOAC or VKA) in 33 patients (12.9%); the main reasons were a favorable course of the cancer in 12 patients (36.4%) and related to patient’s choice in 10 patients (30.3%). The time taken to switch to oral anticoagulation was between 0 and 50 days.

### 3.2. Clinical Outcomes Beyond 6 Months

Between the 6th and 12th month following the index VTE 24 (5.7%), patients experienced a VTE recurrence with a median delay of occurrence of 281 days following the index VTE. The estimated cumulative incidence was 8.0% (95% CI: 4.2%; 15.1%) at 12 months (Table 3, Figure 2). VTE recurrences included PE alone in 12 patients, PE and DVT in 2 patients and DVT alone in 10 patients. Among the 12 DVTs, 4 were lower limb, 1 was upper limb and 7 were central-venous-catheter-related. A total of 16 of 24 patients were receiving an anticoagulant treatment when they experienced a VTE recurrence: these patients had metastatic cancer at inclusion while 13 of them had a cancer in progression at the time of VTE recurrence; 8 were not receiving an anticoagulant treatment when they experienced a VTE recurrence; 5 had metastatic cancer at inclusion and the cancer was in progression at the time of VTE recurrence.

CRB occurred in 21 patients (5.1%) of whom 18 patients were receiving a LMWH while major bleeding occurred in 11 patients (2.7%) of whom 9 patients were receiving a LMWH. Gastrointestinal (GI) clinically relevant bleeding and major bleeding were reported in 9 of 21 patients and in 8 of 11 patients, respectively. In patients with colorectal cancer, GI bleedings represented the majority of clinically relevant (4 of 5) and major bleedings (4 of 5). Between 6 and 12 months the estimated cumulative incidences of clinically relevant bleeding and major bleeding were 4.9% (95% CI: 3.2%; 7.4%) and 2.6% (95% CI: 1.3%; 5.1%), respectively (Table 3, Figure 3).

A total of 96 patients (22.3%) died between the 6th and 12th months following the index VTE, most of the deaths being related to the cancer disease yielding to a cumulative incidence of 30.7% (95% CI: 22.8%; 38.6%).

Estimated cumulative incidences of clinical outcomes were variable according to the type of cancer (Table 3). Cumulative incidences of VTE recurrence in patients with colorectal cancer (12.6%) or lung cancer (13.8%) were higher than in patients with breast cancer (1.5%). Regarding bleeding events, the cumulative incid ence of CRB was higher in patients with colorectal (5.8%) or breast cancer (4.5%) than in patients with lung cancer (1.3%). The same trend was observed for major bleeding. In the same way, estimated cumulative incidences of clinical outcomes were nominally higher in patients with a cancer disease in progression (10.6% of VTE recurrence, 8.8% of CRB and 5.1% of major bleeding) compared to the overall study population (Table 3).

## 4. Discussion

The description of the anticoagulant treatment and the clinical outcomes in USCAT provide additional information on the management of CAT patients up to 12 months after the index event in clinical practice. A total of 432 patients previously treated for 6 months with tinzaparin were available to participate in the USCAT study and the majority of CAT patients (82.5%) were prescribed an anticoagulant treatment beyond 6 months with a significant median treatment duration of approximately 4 to 5 months in addition to the initial 6-month treatment duration, in accordance with clinical practice guidelines (Table 1). Cumulative incidence of VTE recurrence, clinically relevant bleeding and major bleeding was estimated at 8.0% (95% CI 4.2–15.1), 4.9% (95% CI 3.2–7.4) and 2.6% (95% CI 1.3–5.1) respectively. The definitive discontinuation of LMWHs was mostly related to the favorable course of the cancer and the thrombotic disease whereas bleeding risk was a minor concern. The switch from LMWHs to oral anticoagulation between 6 and 12 months was firstly related to the doctor’s decision, then to the patient’s preference. This is consistent with previous observations in France on the management of the anticoagulant treatment in CAT patients, which is based on physician’s decision with a case-by-case approach [18,19].

Our study confirms the influence of the cancer site on the benefit–risk ratio of the anticoagulant treatment beyond 6 months, showing differences in the clinical profile of VTE-related outcomes according to the site of cancer: VTE recurrence was higher in patients with colorectal and lung cancer and major bleeding (including gastrointestinal bleeding) was higher in patients with colorectal cancer. This observation is consistent with the recently reported increased risk of bleeding in patients with gastrointestinal cancers [20,21] and confirms the need of careful precautions when using an anticoagulant treatment in these patients. The influence of the cancer site on the clinical course of VTE during the initial 6-month treatment had also been analyzed in 3,947 cancer patients enrolled in the Registro Informatizado de la Enfermedad TromboEmbólica (RIETE registry) [22]. During course of the anticoagulant therapy, rates of VTE recurrence as rates of major bleeding were higher in patients with lung and colorectal cancer. This strongly suggests the need to develop cancer-specific anticoagulant strategies in the management of patients with CAT. The ongoing prospective double blind API–CAT study (APIxaban Cancer Associated Thrombosis, NCT03692065; https://clinicaltrials.gov/ct2/show/NCT03692065) is the first randomized controlled trial (RCT) to document critical outcomes in CAT patients during the period following 6-month treatment with any anticoagulant. Patients are randomized to receive either apixaban 5mg bid or apixaban 2.5 mg bid with the objective of assessing the optimal long-term dosing regimen. The stratification according to the cancer site may contribute to the tailoring of patient management according to the nature of the underlying malignancy. Considering the profile of patients having experienced a VTE recurrence, our study suggests a close relationship between both the stage and progression of cancer, and the risk of VTE recurrence. These data recommend the extension of the anticoagulant therapy for as long as the cancer is active and/or the patient receives an antineoplastic treatment.

The incidences of clinical outcomes beyond 6 months observed in USCAT are consistent with those observed in post-hoc analyses of randomized controlled trials (RCTs) or observational studies (Figure 4). In the DALTECAN [11] and TiCAT [12] prospective cohort studies, the rates of recurrent VTE beyond 6 months were 4.3% and 1.1% respectively and the rates of major bleeding were 4.3% and 2.7% respectively, close to the 5.7% and 2.7% observed in the USCAT study. In the RCTs, rates of recurrent VTE were 2.9% in the dalteparin group and 1.4% in the edoxaban group of the HOKUSAI study [13], and 4.0% in the rivaroxaban group of the SELECT-D trial [23] while the rates of major bleeding were 1.1% in the dalteparin group and 2.4% in the edoxaban group of the HOKUSAI study [13] and 5.0% in the rivaroxaban group of the SELECT-D trial [23]. In a recently published retrospective registry of 524 CAT patients, anticoagulation was maintained in 222 patients beyond 6 months and up to 24 months in patients with advanced cancer and receiving antineoplastic treatment. In these patients, VTE recurrence and clinically relevant bleeding incidences were 2.7% (95% CI: 1.8; 3.4) and 1.4% (95% CI: 0.7; 2.2), respectively [14].

The absence of randomization in observational studies on the extended anticoagulant treatment means that the evidence provided is limited, drug comparisons are irrelevant, and therefore guidance on the usefulness of the anticoagulant treatment beyond 6 months since the maintenance of the anticoagulant treatment and the choice of the anticoagulant are left to the investigator’s judgment. The 7-to-12-month VTE recurrence raw incidence of 5.7% reported in our study is close to the rate of 4.3% reported in the DALTECAN study [11], while it appears higher than the rate reported in TiCAT [12] and in the LMWH group of the HOKUSAI post-hoc analysis [13] (1.1% and 2.9%, respectively). This difference is possibly be due to differences in patient populations or concomitant antineoplastic treatments. Furthermore, the trend towards the reduction of clinical outcome incidence beyond 6 months compared to the initial 6-month therapy could either be associated to the benefit of the anticoagulant treatment or simply to the natural history of the disease.

Even though the two observational studies discussed in this article were prospective [11,12], USCAT was a retrospective study without randomization and valid control which may represent a limitation regarding missing data. However, of the 487 eligible patients, only 55 patients could not be included in the USCAT study. Furthermore, clinical protocols and event adjudication in PREDICARE [15] and aXa (NCT02898051; https://clinicaltrials.gov/ct2/show/NCT02898051) studies were homogeneous, allowing the inclusion of a large study population in USCAT. Our study is the largest sample size of CAT patients enrolled after 6-month treatment with LMWH (*n* = 432) in an observational study with a relatively large number of patients having completed the 12-month follow-up (*n* = 332). Moreover, patients’ characteristics were documented 6 months after the index VTE i.e., at the time when the treatment strategy for the subsequent 6 months was to be discussed.

A strength of this study is the use of a competitive risk method to estimate outcomes cumulative incidences whereas this method was inappropriately ignored in the other studies. Furthermore, and unlike treatment groups in RCTs previously discussed, there was no ambivalence clause.

## 5. Conclusions

Clinical practice guidelines suggest the extension of the anticoagulant treatment beyond 6 months in CAT patients with active cancer, or receiving an antineoplastic therapy, while LMWH and DOACs are recommended in this setting. However, these recommendations are not supported by robust data from randomized trials and the therapeutic strategy in clinical practice is left to the physician’s judgment on a case-by-case basis. The identification in our study of cancer types particularly associated to VTE recurrence or bleeding is likely to provide useful guidance for the management of the anticoagulant treatment in patients with CAT. On this basis, further clinical research on the management of patients with CAT beyond 6 months based on the type of cancer is warranted.

## Figures and Tables

**Figure 1 cancers-12-02256-f001:**
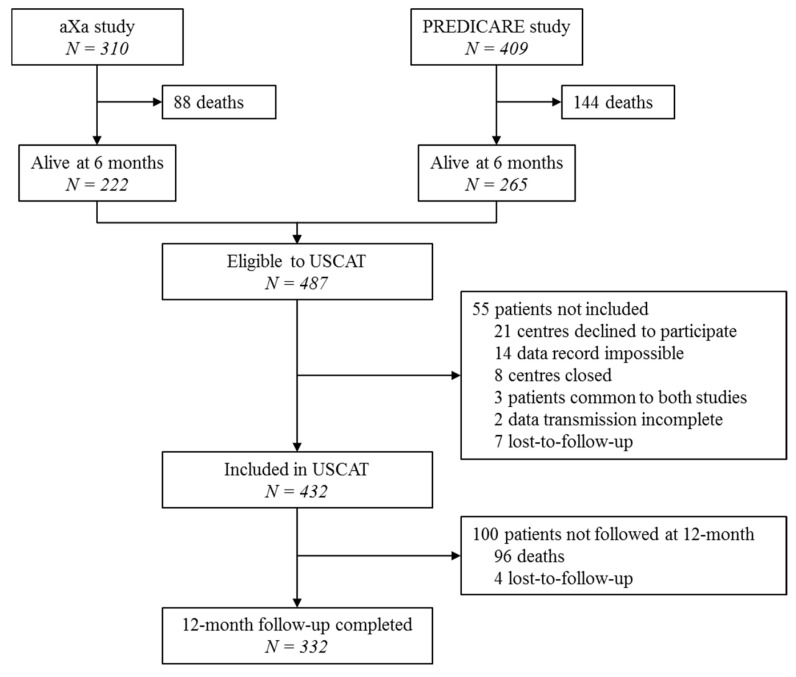
Patient flow chart in the USCAT (USual care of Cancer Associated Thrombosis) study.

**Figure 2 cancers-12-02256-f002:**
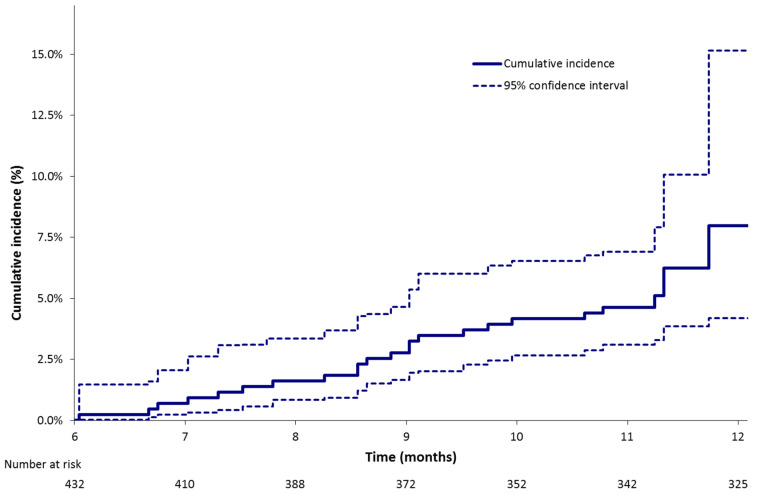
Estimated cumulative incidence of VTE recurrence between 6 and 12 months following the index event (Kalbfleisch and Prentice method).

**Figure 3 cancers-12-02256-f003:**
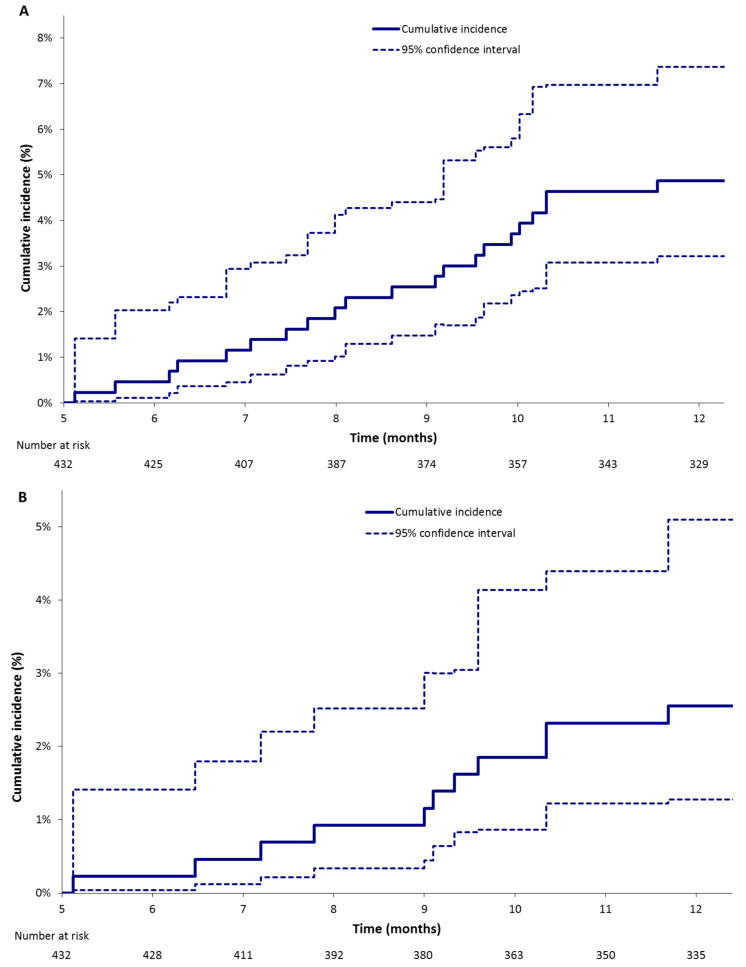
Estimated cumulative incidences of (**A**) clinically relevant bleeding, sum of major bleeding and clinically relevant non major bleeding and (**B**) major bleeding between 6 and 12 months following the index event (Kalbfleisch and Prentice method).

**Figure 4 cancers-12-02256-f004:**
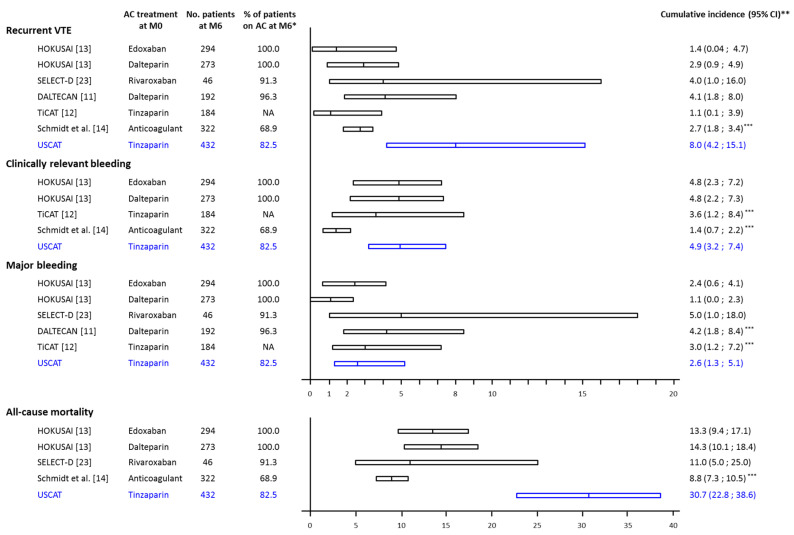
Estimated cumulative incidence of clinical outcomes during the 7–12-month study period in treated CAT patients. AC: anticoagulant, No. pts: number of patients, MO: VTE diagnosis, M6: 6-month from VTE diagnosis; * percentage of patients on anticoagulant treatment at the beginning of the period from 6- to 12- month; ** Cumulative incidence estimated by the Kaplan-Meier method except for Schmidt et al., and USCAT studies where cumulated incidence of recurrent VTE, clinically relevant bleeding and major bleeding were estimated by the Kalbfleisch and Prentice method taking into account the competing risk of death; *** Standardized on 6 months.

**Table 1 cancers-12-02256-t001:** International guidelines for the long-term treatment of cancer-associated thrombosis.

International Guidelines	Long-Term Anticoagulation	Treatment Duration
ACCP [5]	LMWH over VKA (Grade 2B), dabigatran (Grade 2C), rivaroxaban (Grade 2C), apixaban (Grade 2C), or edoxaban (Grade 2C)	>3 months in patients at low bleeding risk (1B) or high risk of bleeding (2B)
ASCO [6]	LMWH, edoxaban, or rivaroxaban for at least 6 months are preferred because of improved efficacy over vitamin K antagonists (VKA). Caution with DOAC warranted in settings with high risk for mucosal bleeding. Drug-drug interaction should be checked prior to using a DOAC (strong)	Anticoagulation with LMWH, DOAC, or VKA beyond the initial 6 months should be offered to select patients with active cancer, such as those with metastatic disease or those receiving chemotherapy (weak to moderate).
ACI FORUM [9]	LMWH	At least 6 months Stop treatment in the absence of tumor activity of antineoplastic treatment On the contrary, maintain treatment beyond 6 months with periodic re-evaluation of benefit/risk ratio
ITAC-CME [4]	LMWH preferred to VKA (1A) DOAC if CrCl ≥ 30 mL/min, in the absence of drug-drug interactions or insufficient digestive absorption (1A) Precaution if GI cancer	At least 6 months (1A) Beyond 6 months discuss prolongation case by case based on the analysis of benefit/risk ratio (Guidance)
NCCN [10]	LMWH recommended Category 1 recommendations for both edoxaban and apixaban VKA possible	At least 3 months Maintain treatment as long cancer is active, anti-neoplastic treatment maintained or increased risk of VTE recurrence (2A)
IFS [7]	LMWH recommended (1+) DOAC if LMWH not tolerated (2+)	At least 6 months Maintain treatment beyond 6 months as long cancer is active, based on patient’s preference, bleeding risk: LMWH (2+), VKA (2+) or DOAC (2+) full treatment dose

ACCP: American College of Chest Physicians; ASCO: American Society of Clinical Oncology; ITAC-CME: International Initiative on Thrombosis and Cancer; NCCN: National Comprehensive Cancer Network; ISF: Inter French Societies; LMWH: low-molecular-weight heparin; VKA: vitamin K antagonist; DOAC: direct oral anticoagulant; VTE: venous thromboembolism.

**Table 2 cancers-12-02256-t002:** Baseline demographic and clinical characteristics of the 432 included patients.

Patient Characteristics	All Patients (*n* = 432)
Mean age (years) ± SD	66.5 ± 12.7
Age ≥ 75 years, no. (%)	128 (29.6)
Male sex, no. (%)	207 (47.9)
**Site of cancer disease, no. (%)**	
**Solid tumour**	394 (91.2)
Colorectal	85 (21.6)
Lung	79 (20.1)
Breast	66 (16.8)
Genitourinary	62 (15.7)
Gynaecologic	42 (10.7)
Pancreas	14 (3.6)
Upper gastrointestinal	13 (3.0)
Hepatobiliary	10 (2.5)
Other	23 (5.3)
**Haematologic tumour**	30 (6.9)
Non-Hodgkin lymphoma	14 (46.7)
Multiple myeloma	7 (23.3)
Leukaemia	7 (23.3)
Hodgkin lymphoma	2 (6.7)
Other	8 (1.9)
**Stage (*n* = 390), no. (%)**	
1	45 (11.5)
2	25 (6.4)
3 or 4	320 (82.1)
**Cancer evolution (*n* = 424), no. (%)**	
Remission	66 (15.6)
Stability	141 (33.3)
Progression	217 (51.2)
**Index VTE *, no. (%)**	
PE ± DVT	318 (73.6)
DVT alone	114 (26.4)
-Proximal DVT	70
-Distal DVT	54
-Unclear	2
**Anticoagulant treatment (*n* = 422), no. (%)**	
Stopped before the end of the initial 6-month treatment period	60 (14.2)
Stopped at the end of the initial 6-month treatment period	14 (3.3)
Maintained at 6 months	348 (82.5)
LMWH	256 (73.6)
VKA	56 (16.1)
DOAC	30 (8.6)
UFH	3 (0.9)
Fondaparinux	3 (0.9)

SD: standard deviation; VTE: venous thromboembolism; PE: pulmonary embolism; DVT: deep vein thrombosis; LMWH: low-molecular-weight heparin; VKA: vitamin K antagonist; DOAC: direct oral anticoagulant; UFH: unfractionated heparin; * more than one event in several patients.

**Table 3 cancers-12-02256-t003:** Estimated cumulative incidence of clinical outcomes during the 7–12-month period following the index event [% (95% CI)] in USCAT.

Clinical Outcomes	All Patients *n* = 432 *	According to the Type of Cancer				**According to the State of Cancer**	
		Colorectal *n* = 85	Lung *n* = 79	Breast *n* = 66	Other *n* = 202	Cancer Progression *n* = 217	Metastatic Cancer *n* = 320
VTE recurrence¶	8.0 (4.2; 15.1)	12.6 (4.6; 34.3)	13.8 (8.4; 22.8)	1.5 (0.3; 8.4)	3.5 (1.7; 7.0)	10.6 (5.3; 21.2)	8.7 (5.1; 14.9)
CRB **¶	4.9 (3.2; 7.4)	5.8 (2.5; 13.6)	1.3 (0.2; 9.1)	4.5 (1.5; 13.8)	6.0 (3.4; 10.4)	8.8 (5.6; 13.7)	5.3 (3.2; 8.8)
Major bleeding¶	2.6 (1.3; 5.1)	5.8 (2.6; 13.2)	0 (0.0)	1.5 (0.2; 12.1)	2.5 (1.2; 5.3)	5.1 (2.8; 9.1)	2.8 (1.4; 5.6)
Deaths §	30.7 (22.8; 38.6)	24.2 (11.4; 37.1)	42.2 (25.0; 59.4)	15.2 (6.5; 23.8)	34.5 (19.9; 49.1)	52.9 (41.0; 64.8)	36.7 (27.6; 45.7)

* VTE recurrence, bleeding events and deaths were documented in 418, 415 and 430 patients, respectively; ** CRB: clinically relevant bleeding, sum of major bleeding and clinically relevant non major bleeding; ¶ Kalbfleisch and Prentice; § Kaplan Meier.

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
