# Peer review of "Long-Term Treatment of Cancer-Associated Thrombosis (CAT) Beyond 6 Months in the Medical Practice: USCAT, a 432-Patient Retrospective Non-Interventional Study"

_cancers, 2020, doi:10.3390/cancers12082256_

Round 1

Reviewer 1 Report

The manuscript entitled "Long-Term Treatment of Cancer-associated 1 Thrombosis (CAT) beyond 6 months in the medical 2 practice: USCAT, a 432-patient retrospective non-3 interventional study." by Mahé, et al. suggest a cancer type-based management of anticoagulant therapy after development of VTE.

In this study, VTE recurrence was higher in colorectal and lung cancer compared with other cancer types. I agree with the authors proposal to lead tailor patient management; however, would like to encourage the authors to describe about the mechanism of recurrence of VTE despite several anticoagulant agents were used. So please provide the authors opinion in the Discussion section regarding differences in tumor environment or tumor stage or metastatic pattern among the cancer types.

Another query is how long anticoagulant should be continued for preventing VTE with or without chemotherapy when there is no risk of bleeding. Please state your opinion in the discussion.

Author Response

We thank the reviewer for his comments that contribute to improve the manuscript.

Data about cancer status of patients experiencing a VTE recurrence were added (L172): Sixteen of 24 patients were receiving an anticoagulant treatment when they experienced a VTE recurrence: these patients had metastatic cancer at inclusion while 13 of them had a cancer in progression at the time of VTE recurrence; 8 were not receiving an anticoagulant treatment when they experienced a VTE recurrence ; 5 had metastatic cancer at inclusion and the cancer was in progression at the time of VTE recurrence.

A comment has been inserted in the discussion section (L219): Considering the profile of patients having experienced a VTE recurrence, our study suggests a close relationship between both the stage and progression of cancer, and the risk of VTE recurrence. These data support recommend the extension of the anticoagulant therapy for as long as the cancer is active and/or the patient receives an antineoplastic treatment.

Overall, English language and style have been reviewed and improved when needed.

Reviewer 2 Report

The study by Mahé provides some useful information regarding the long-term treatment of patients with cancer-associated thrombosis (CAT). I have the following comments:

The numbers of study participants included in USCAT is highly confusing. Obviously, many clinical data were not available for all patients, and the authors use the term "documented patients" to account for these discrepancies. The numbers of patients, for which a specific information was available or not, must be presented more clearly in Table 1. Still, some numbers appear incorrect. For instance, page 4, Results: The index VTE was a PE (+/- DVT) in 318 patients (74 %) and isolated DVT in 112 patients (26 %), which sums up to 430 and not to 432 patients. Furthermore, the numbers of isolated PE and PE combined with a DVT sum up to 317 and not to 318 patients. The authors need to carefully check their numbers throughout the manuscript.

Page 5, lines 153 to 156: This section is confusing with regard to median treatment durations in patients switched to a VKA or DOAC. Was the median total treatment duration after the index VTE event in patients switched to a DAOC only one day longer than the median treatment duration in USCAT?          

In the main body of the manuscript, all abbreviations should be spelled out first, e.g. CAT (page 2, line 61). The abbreviation USCAT should be first introduced in the Materials and Methods section, and not in the Results section.

Page 2, line 64: I do not agree that clinical practice guidelines recommend a minimum of 6 months of anticoagulation in patients with CAT. In Table 1, the authors themselves cite guidelines recommending > 3 months of anticoagulation. I suggest changing "a minimum of 6 months" to "a minimum of 3–6 months".

Table 1: The NCCN guideline has been updated in 2020 with category 1 recommendations for both edoxaban and apixaban. This needs to be changed.

Have results of the aXa study not been reported? The status of this study needs to be clarified. Also, please specify how patients were followed up in USCAT. Although this is a retrospective study, the authors should comment on this issue.

Page 3, line 104: I would change the sentence to "1) the incidence of recurrent VTE events confirmed by appropriate imaging studies", since ultrasonography is not a radiographic examination.

Page 3, line 108: Change Hemostasis (in ISTH) to Haemostasis.

Page 3, line 9: The ISTH definition of major bleeding does not require symptomatic, but overt bleeding.       

Page 9, lines 198 to 200, and Table 3: What were event rates for safety and efficacy outcomes in patients with cancer progression or metastatic cancer referred to?

Page 11, Conclusion: The statement that LMWH remains the main choice in patients with active cancer or receiving antineoplastic therapy should be toned down in light of recent clinical trial findings and guideline recommendations.

Author Response

We thank the reviewer for his comments that contribute to improve the manuscript.

The number of patients with PE and DVT has been checked and corrected in the text and the Table 2.

Page 5, lines 153 to 156: Indeed the median total treatment duration after the index VTE event in patients switched to a DOAC was only one day longer than the median treatment duration in USCAT since most DOACs were prescribed 6 months after the index event.

Abbreviations have been appropriately spelled out.

Page 2: “a minimum of 6 months” changed to “a minimum of 3-6 months”

Table 1: NCCN updated to 2020 version

The results of the AXA study have not been reported as yet.

The USCAT study is a retrospective cohort with only a collection of medical data. No additional medical procedures beyond the usual intake were performed as part of the study. In accordance with French regulations (loi n° 2012-300 du 5 mars 2012 relative aux recherches impliquant la personne humaine (Jardé law), the informed consent of the patient is not required. An information letter has been sent to patients to inform them of their rights regarding the protection of their data in accordance with the regulations.

As mentioned L 128, the follow-up has been performed as follows: “The same independent investigator had access to the primary data and reviewed medical records of all included patients and recorded data in a Standardized case report form”

Page 9 L187: Events rates for safety and efficacy (CRB, major bleeding and VTE recurrence) have been described in patients with cancer disease in progression: “estimated cumulative incidences of clinical outcomes were nominally higher in patients with a cancer disease in progression (10.6% of VTE recurrence, 8.8% of CRB and 5.1% of major bleeding) compared to the overall study population (Table 3)”. Additional descriptive data have been provided in the description of patients with VTE recurrence (L173).

Conclusion: the statement is modified in: “Clinical practice guidelines suggest the extension of the anticoagulant treatment beyond 6 months in CAT patients with active cancer or receiving an antineoplastic therapy while LMWH and DOACs are recommended in this setting”.

Overall, English language and style have been reviewed and improved when needed.

Reviewer 3 Report

The research questions raised by the authors are interesting. Should we continue anticoagulation beyond 6 months in cancer patients with VTE? The retrospective nature of this study, do not answer this specific question.The study do not have a valid control. Not clear why some patients stopped anticoagulation at 6 months, while other continued treatment more.   

Author Response

The USCAT study suffers the limitations of the research design of all retrospective observational studies. We mention in the discussion L 244 that “USCAT was a retrospective study without randomization and valid control which may represent a limitation regarding missing data”. The interest of such studies is that they provide a picture of the clinical practice in the real life.

We stated (L 160) that the main reasons for treatment discontinuation “were a favorable course of the cancer (16 patients, 18.6%) and VTE disease (11 patients, 12.8%), whereas concern about bleeding risk was low (2 patients, 2.3%)”.

Overall, English language and style have been reviewed and improved when needed.

Round 2

Reviewer 2 Report

The authors have adequately addressed my concerns. 

Reviewer 3 Report

The study is retrospective though the results are interesting.